# Differentially Private Optimization on Large Model at Small Cost

## Abstract

Differentially private (DP) optimization is the standard paradigm to learn large neural networks that are accurate and privacy-preserving. The computational cost for DP deep learning, however, is notoriously heavy due to the per-sample gradient clipping. Existing DP implementations are $2 - 1000\times$ more costly in time and space complexity than the standard (non-private) training. In this work, we develop a novel Book-Keeping (BK) technique that implements existing DP optimizers (thus achieving the same accuracy), with a substantial improvement on the computational cost. Specifically, BK enables DP training on large models and high dimensional data to be roughly as efficient as the standard training, whereas previous DP algorithms can be inefficient or incapable of training due to memory error. The computational advantage of BK is supported by the complexity analysis as well as extensive experiments on vision and language tasks. Our implementation achieves state-of-the-art (SOTA) accuracy with very small extra cost: on GPT2 and at almost the same memory cost ($< 1\%$ overhead), BK has $1.03\times$ the time complexity of the standard training ($0.83\times$ training speed in practice), and $0.61\times$ the time complexity of the most efficient DP implementation ($1.36\times$ training speed in practice). We will open-source the codebase for the BK algorithm.

## 1 Introduction

Deep learning with differential privacy (DP; Dwork et al. (2006)) has shown strong performance while guaranteeing rigorous protection against privacy risks, especially on large models that tend to memorize and leak the training data Carlini et al. (2021); Haim et al. (2022); Shokri et al. (2017). For example, recent advances have shed light on the success of DP GPT2 Li et al. (2021); Bu et al. (2022b); Yu et al. (2021), which achieves $64.6$ BLEU score[1] at strong privacy guarantee ($\epsilon = 3$), on the text generation task using E2E restaurant review dataset. This is only marginally below the standard non-private GPT2 (BLEU score 66.8). Similarly, on computer vision tasks ($\epsilon = 2$), DP vision transformers and ResNets have obtained $97.1\%/86.2\%$ accuracy on CIFAR10/100 by Bu et al. (2022a) and over $81\%$ accuracy on ImageNet by De et al. (2022); Mehta et al. (2022).

However, DP training of large neural networks is well-known to be computationally burdensome in comparison to the standard training, in terms of both the training time and the memory cost. For instance, training a small recurrent neural network (0.598M parameters) experiences a $1000\times$ slowdown using DP optimizers in Tensorflow-Privacy (TF-Privacy) library Bu et al. (2021), and training a small convolutional neural network (CNN, 0.605M parameters) on CIFAR10 has a $24\times$ slowdown with Tensorflow 2 and the XLA compiler Subramani et al. (2021). Even with SOTA efficient implementations, large models such as RoBERTa Liu et al. (2019), GPT2 Radford et al. (2019), ResNet He et al. (2016), VGG Simonyan & Zisserman (2014), ViT Dosovitskiy et al. (2020) and its variants, experience about $2 - 3\times$ slowdown in Pytorch Li et al. (2021); Bu et al. (2022a) and $2 - 9\times$ slowdown in JAX Kurakin et al. (2022); De et al. (2022), with possibly $4 - 20\times$ memory overhead Bu et al. (2022a); Li et al. (2021); Subramani et al. (2021) if not out of memory.

The efficiency bottleneck in DP deep learning lies in the per-sample gradient clipping, which restricts the magnitude of each per-sample gradient in the mini-batch. Applying the clipping jointly with the

---

[1] BLEU (BiLingual Evaluation Understudy) is a metric (0-100) for automatically evaluating translated text. BLEU > 60 is considered as "very high quality, adequate, and fluent translations, often better than human".

Gaussian noise addition, one can privately release the gradient to arbitrary optimizers like SGD and Adam, and thus guarantee the privacy of the training as described in Section 1.3:

$$\text{private gradient:} \quad \hat{\mathbf{G}} := \sum_i \boldsymbol{g}_i \cdot C(\|\boldsymbol{g}_i\|_2) + \sigma_{\text{DP}} \cdot \mathcal{N}(0, \mathbf{I}),$$
$$\text{private optimizer (e.g. SGD):} \quad \mathbf{w}_{t+1} = \mathbf{w}_t - \eta \hat{\mathbf{G}}. \tag{1}$$

Here $\mathbf{w}$ is the model parameters, $\mathcal{L}_i$ is the per-sample loss, $\boldsymbol{g}_i = \frac{\partial \mathcal{L}_i}{\partial \mathbf{W}}$ is the per-sample gradient, $\eta$ is the learning rate, $\sigma_{\text{DP}}$ is the noise magnitude that defines the privacy loss, and $C(\|\boldsymbol{g}_i\|)$ or simply $C_i$ is the per-sample clipping factor, e.g. $\min\{R/\|\boldsymbol{g}_i\|, 1\}$ in Abadi et al. (2016), with a clipping threshold $R$.

At high level, previous work have tackled the efficiency bottleneck with various approaches.

$$\underbrace{\text{I. optimizer}}_{\text{DP-SGD, DP-Adam}} - - - - \underbrace{\text{II. parameter efficiency}}_{\text{last layer, LoRA, Adapter}} - - - - \underbrace{\text{III. implementation}}_{\text{Opacus, GhostClip, BK}} - - - - \underbrace{\text{IV. platform}}_{\text{Pytorch, JAX, TF}}$$

One approach (part II) focuses on the parameter efficiency by partially training a neural network, in contrast to full fine-tuning all model parameters, e.g. only the last output layer Tramer & Boneh (2020), the adapter layers Houlsby et al. (2019); Mahabadi et al. (2021), or the Low-Rank Adaptation (LoRA) Hu et al. (2021); Yu et al. (2021). For example, Mehta et al. (2022) accelerate the DP training on ImageNet Deng et al. (2009) up to $30\times$ by only training the last layer of ResNet152. Noticeably, parameter efficient fine-tuning does not improve on the efficiency in terms of complexity per parameter, rather than reducing the number of parameters. Furthermore, this approach oftentimes leads to some accuracy degradation compared to DP full fine-tuning Bu et al. (2020); Mehta et al. (2022); Li et al. (2021); Yu et al. (2021).

An orthogonal approach, including this work, focuses on the computation efficiency (part III), i.e. reducing the time and space complexity through efficient implementations, without affecting the DP optimizers (part I) and thus their performance. We will elaborate on multiple methods in Section 1.2. Additionally, these methods can be compiled on different platforms (part IV) such as Tensorflow 2(XLA), JAX and Pytorch Li et al. (2021); Subramani et al. (2021); De et al. (2022); Kurakin et al. (2022), where remarkable speed difference has been observed in some cases, even with the same implementation. For example, Subramani et al. (2021) implemented DP-SGD using JAX and claimed its efficiency advantage over the same algorithm using Tensorflow or Pytorch.

## 1.1 CONTRIBUTIONS

1. **[Algorithm]** We propose the book-keeping (BK) algorithm that makes existing DP optimizers fast and memory efficient, especially comparable to non-private optimizers. We demonstrate BK via the computation graph in Figure 1. The highlight is that BK *only uses one back-propagation* and *never instantiates per-sample gradients* $\{\frac{\partial \mathcal{L}_i}{\partial \mathbf{W}}\}_{i=1}^B$.

2. **[Analysis]** We analyze the complexity to show that *BK has almost the same time and space complexity as non-DP training*, especially when the feature dimension is small (see Table 5).

3. **[Extension]** We strengthen BK using a layerwise decision to mix with Opacus (see Section 3.2), which proves to be efficient when the feature dimension is large (and difficult for GhostClip). We also extend BK to the parameter efficient fine-tuning such as DP LoRA and Adapter.

4. **[Codebase]** We develop a Pytorch (Paszke et al., 2019) codebase for our BK algorithm, leveraging the auto-differentiation technique on the computation graph and a new trick in Appendix D.2.

5. **[Experiments]** We demonstrate the amazing efficiency of BK on training large models, saving the memory up to $10\times$ and boosting the speed by $30\% - 5\times$ than previous DP implementations.

| Dataset | SOTA setting | Model | Time /Epoch | Relative Speed over GhostClip | over Opacus | over non-DP |
|---------|--------------|-------|-------------|-------------------------------|-------------|-------------|
| QQP | Li et al. (2021) | RoBERTa-large (355M) | 70:04 | $1.36\times$ | $1.96\times$ | $0.77\times (0.89\times)$ |
| E2E | Li et al. (2021) | GPT2-large (774M) | 10:01 | $1.36\times$ | $4.41\times$ | $0.83\times (0.97\times)$ |
| CIFAR | Bu et al. (2022a) | BEiT-large (304M) | 6:35 | $1.33\times$ | $38.3\times$ | $0.76\times (0.92\times)$ |

Table 1: A preview of BK's efficiency on DP tasks (complexity in orange; extended in Table 9).

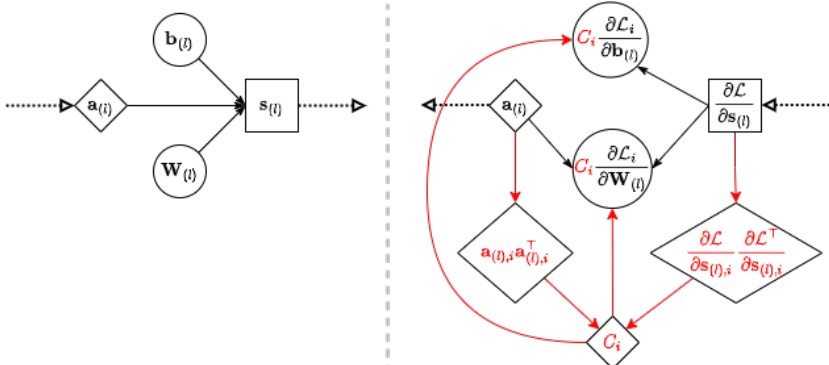

Figure 1: Forward pass and back-propagation of the $l$-th linear layer (standard training is in black; DP training by our book-keeping algorithm is added in red). Here $\boldsymbol{a}_{(l)}$ is the activation tensor, $\boldsymbol{s}_{(l)}$ is the layer output, $\mathbf{W}_{(l)}, \mathbf{b}_{(l)}$ are weight and bias, $\mathcal{L}_i, \mathcal{L}$ are the per-sample loss and the summed loss. The dotted arrow represents the inter-layer operation such as activation, pooling, or normalization.

## 1.2 RELATED WORKS

Previous arts have developed different implementations of the same DP optimizer in Equation (1). TF-Privacy Tensorflow back-propagates a vectorized loss $[\mathcal{L}_1, \cdots, \mathcal{L}_B]$ to compute the per-sample gradients, each from one back-propagation, which is memory-efficient but slow. Opacus Yousefpour et al. (2021) and Rochette et al. (2019) accelerate the training significantly using the outer product trick in Goodfellow et al. (2014), though incurring heavy memory burden so as to store the per-sample gradients. This memory burden is partially alleviated in FastGradClip Lee & Kifer (2020) by sharing the space complexity in two rounds of back-propagation, hence almost doubling the time complexity. In ghost clipping Goodfellow (2015); Li et al. (2021); Bu et al. (2022a), the per-sample gradients can be clipped without being instantiated, thus both time and space complexity can be further improved if the feature dimension is small. We refer interested readers to Figure 3 and Appendix C for algorithmic details of these implementations.

We now compare BK to different implementations in Table 2 and Figure 2. In what follows, $B$ is the batch size[2], $T_{(l)}$ is the feature dimension[3], $d_{(l)}, p_{(l)}$ are the input or output dimension of a layer.

| | Non-DP | TF-privacy | Opacus | FastGradClip | GhostClip | BK (ours) |
|---|---|---|---|---|---|---|
| Instantiating per-sample grad | ✗ | ✓ | ✓ | ✓ | ✗ | ✗ |
| Storing every layer's grad | ✗ | ✗ | ✓ | ✗ | ✗ | ✗ |
| Instantiating non-DP grad | ✓ | ✓ | ✓ | ✗ | ✓ | ✗ |
| Number of back-propagation | 1 | $B$ | 1 | 2 | 2 | 1 |
| Time Complexity of Clipping | $6BTpd$ | $6BTpd$ | $8BTpd$ | $8BTpd$ | $10BTpd + O(BT^2)$ | $\approx 6BTpd$ |
| Memory Overhead to non-DP | 0 | 0 | $Bpd$ | $Bpd$ | $2BT^2$ | $\min\{2BT^2, BTpd\}$ |
| Scalable to large model | ✓ | ✗ | ✗ | ✗ | ✓ | ✓ |
| Scalable to high-dim input | ✓ | ✗ | ✓ | ✓ | ✗ | ✓ |

Table 2: Summary of different DP implementations on a linear/convolution layer $\mathbb{R}^{B \times T_{(l)} \times d_{(l)}} \rightarrow \mathbb{R}^{B \times T_{(l)} \times p_{(l)}}$. The main bottleneck is marked in red.

## 1.3 PRELIMINARIES

We work with the $(\epsilon, \delta)$-DP by Dwork et al. (2006), which makes it difficult for any privacy attacker to distinguish or detect an arbitrary training sample, even with full access to the model (see Appendix A for details). In deep learning, DP is achieved by training on the private gradient in Equation (1) with any optimizer such as SGD, Adam, FedAvg, etc. Essentially, the private gradient is the addition of Gaussian noise to the sum of clipped per-sample gradients, which guarantees the DP protection through the privacy accounting theorems Abadi et al. (2016); Mironov (2017); Dong et al. (2019); Zhu et al. (2021); Gopi et al. (2021); Koskela et al. (2020).

---

[2]We report the physical batch size, which affects the efficiency; the accuracy is only affected by the logical batch size, which can be implemented through the gradient accumulation of physical batch size.

[3]For non-sequential data, $T = 1$; for texts, $T$ is the sequence length, which is layer-independent; for images (or videos), $T_{(l)}$ is the height×width(×time) of hidden feature representation, which is layer-dependent.

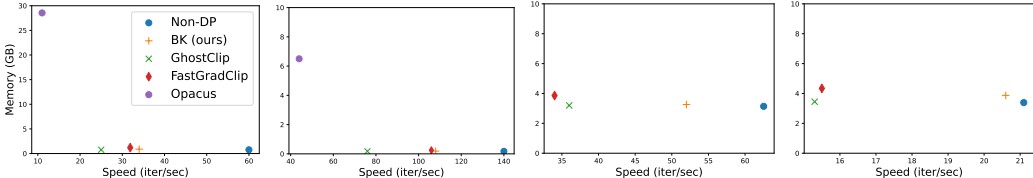

Figure 2: Speed and memory on MLP and CIFAR100 (images are flattened into vectors). Left to right: deep network (50 layers, width 1000, 50M parameters, batch size 128), shallow network (10 layers, width 1000, 10M parameters, batch size 128), and wide network (10 layers, width 5000, 250M parameters, batch size 128 or 1024; Opacus is OOM). See more ablation study in Appendix F.

## 2    BOOK-KEEPING: EFFICIENT DP TRAINING IN LOW DIMENSION

The main computational bottleneck of DP training comes from the per-sample gradient clipping, or from the computation of per-sample gradient norms, to be exact. One widely used approach in Opacus, TF-privacy, and FastGradClip, is to instantiate the per-sample gradients and then deriving their norms. Straight-forward implementation of this approach on a mini-batch of per-sample losses requires $B$ rounds of back-propagation (unacceptable slowdown) or $B\times$ gradient storage (unacceptable memory burden; see Opacus in Figure 2). Consequently, these implementations are not suitable for large model training. For instance, Li et al. (2021) shows that, when training GPT2-large (774M parameters), Opacus Yousefpour et al. (2021) and JAX Subramani et al. (2021) cannot fit even one single sample into a 24GB GPU.

An alternative approach, termed as the ghost clipping (GhostClip), directly computes the per-sample gradient norms without computing the gradients themselves. This is made possible, unfortunately, through two rounds of back-propagation. During the first back-propagation, one uses the regular loss $\sum_i \mathcal{L}_i$ and extracts the activation tensor and the output gradient $(\boldsymbol{a}, \frac{\partial \mathcal{L}}{\partial \boldsymbol{s}})$. One can use an algebraic trick in Equation (2) to compute the per-sample gradient norms $\{\|\frac{\partial \mathcal{L}_i}{\partial \mathbf{W}}\|\}_i$ and the clipping factors $\{C_i\}_i$ in Equation (1). During the second back-propagation, one uses the reweighted loss $\sum_i C_i \mathcal{L}_i$ whose gradient is directly the weighted gradient $\sum_i C_i \boldsymbol{g}_i$, which constitutes the private gradient we need. Note that this double back-propagation roughly doubles the training time (or to be more precise, $10/6 \approx 1.667\times$ when $T$ is small; see Table 2).

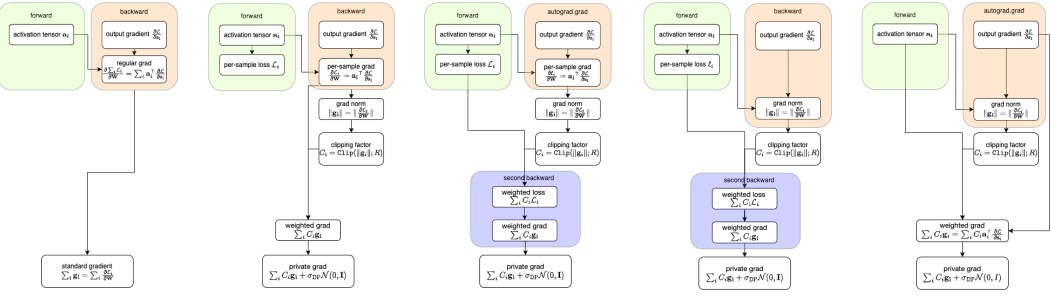

Figure 3: Standard (non-DP), Opacus, FastGradClip, GhostClip, BK implementations. Notice that BK learns to directly compute weighted gradient from Opacus, to compute the ghost norm from GhostClip, to use auto-differentiation instead of full back-propagation from FastGradClip.

To make the DP training as efficient as the standard training, we propose the book-keeping technique (BK) that $\langle 1 \rangle$ only requires a single back-propagation, like Opacus and standard training; $\langle 2 \rangle$ does not instantiate the per-sample gradients, like GhostClip.

### 2.1    BOOK-KEEPING ALGORITHMS

BK algorithms in their base forms are built on GhostClip and especially the *ghost norm* trick, so as to avoid instantiating the memory costly per-sample gradients: as can be seen in Algorithm 1 and Figure 3, $\frac{\partial \mathcal{L}_i}{\partial \mathbf{W}} = \boldsymbol{a}_i^\top \frac{\partial \mathcal{L}}{\partial \boldsymbol{s}_i}$ is not computed throughout the training. In comparison to GhostClip, our

significant improvement is solely on the speed (see Table 2) through two novel tricks: the *book-keeping* and the *ghost differentiation*. The entire BK algorithm is built on the understanding of computation graph in Appendix A. Note that these tricks also offer improved efficiency for existing implementations, to be presented in Section 2.4. We now elaborate on these tricks.

$$\text{BK (base)} = \underbrace{\text{ghost norm}}_{\text{from GhostClip}} + \underbrace{\text{book-keeping}}_{\text{ours}} + \underbrace{\text{ghost differentiation}}_{\text{ours}}$$

---

**Algorithm 1** Differentially private deep learning with BK algorithm (one iteration)

---

**Parameters:** $l$-th layer weights $\mathbf{W}_{(l)}$, number of layers $L$, noise level $\sigma$.

1: **for** layer $l \in 1, 2, \cdots, L$ **do**
2:      Get activation tensor $\{\boldsymbol{a}_{(l),i}\}$ by Pytorch forward hook
3: **for** layer $l \in L, L-1, \cdots, 1$ **do**
4:      Get output gradient $\{\frac{\partial \mathcal{L}}{\partial \boldsymbol{s}_{(l),i}}\}$ by Pytorch backward hook
5:      Compute per-example gradient norm $\|\frac{\partial \mathcal{L}_i}{\partial \mathbf{W}_{(l)}}\|_F^2$ by ghost norm trick in Equation (2)
6: Aggregate gradient norm across all layers: $\|\frac{\partial \mathcal{L}_i}{\partial \mathbf{W}}\|_F^2 = \sum_l \|\frac{\partial \mathcal{L}_i}{\partial \mathbf{W}_{(l)}}\|_F^2$
7: Compute clipping factor: $C_i = C(\|\frac{\partial \mathcal{L}_i}{\partial \mathbf{W}}\|_F; R)$
8: **for** layer $l \in L, L-1, \cdots, 1$ **do**
9:      Compute sum of clipped gradients $\mathbf{G}_l = \boldsymbol{a}_{(l)}^\top \text{diag}(C_1, C_2, \cdots) \frac{\partial \mathcal{L}}{\partial \boldsymbol{s}_{(l)}}$
10:      Delete $\{\boldsymbol{a}_{(l),i}\}, \{\frac{\partial \mathcal{L}}{\partial \boldsymbol{s}_{(l),i}}\}$
11: Add Gaussian noise $\hat{\mathbf{G}} = \mathbf{G} + \sigma R \cdot \mathcal{N}(0, \mathbf{I})$
12: Apply SGD/Adam/LAMB with the private gradient $\hat{\mathbf{G}}$ on $\mathbf{W}$

---

**Ghost norm trick** The ghost norm trick Goodfellow (2015) computes the gradient norm without the gradient: while the gradient is instantiated by the multiplication in Equation (2), the gradient norm can be computed without $\boldsymbol{a}_i$ meeting $\frac{\partial \mathcal{L}}{\partial \boldsymbol{s}_i}$. This is applicable to generalized linear layers including the linear, the embedding Li et al. (2021), and the convolution layers Bu et al. (2022a). We demonstrate this trick using a simple linear layer $\boldsymbol{s}_i = \boldsymbol{a}_i \mathbf{W}$, where $\mathbf{W} \in \mathbb{R}^{d \times p}$ is the weight matrix, $\boldsymbol{a} \in \mathbb{R}^{B \times T \times d}$ is the mini-batch input of this layer (a.k.a. the activation tensor) and $\boldsymbol{s} \in \mathbb{R}^{B \times T \times p}$ is the output. Given that the output gradient $\frac{\partial \mathcal{L}}{\partial \boldsymbol{s}}$ is readily available in the back-propagation, for DP and standard training, one can directly derive the per-sample gradient norm

$$\left\|\frac{\partial \mathcal{L}_i}{\partial \mathbf{W}}\right\|_{\text{Frobenius}}^2 = \text{vec}\left(\frac{\partial \mathcal{L}}{\partial \boldsymbol{s}_i} \frac{\partial \mathcal{L}}{\partial \boldsymbol{s}_i}^\top\right) \cdot \text{vec}\left(\boldsymbol{a}_i \boldsymbol{a}_i^\top\right) \text{ without computing } \frac{\partial \mathcal{L}_i}{\partial \mathbf{W}} = \boldsymbol{a}_i^\top \frac{\partial \mathcal{L}}{\partial \boldsymbol{s}_i}. \quad (2)$$

Here 'vec' means flattening the $T \times T$ matrix to a vector. This trick is particularly efficient when $T$ is small, reducing the space complexity from $O(Bpd)$ to $O(BT^2)$ by Table 3.

**Book-keeping trick** This trick improves the time complexity by removing the second back-propagation from GhostClip. Our idea is to book-keep and re-use the output gradient $\frac{\partial \mathcal{L}}{\partial \boldsymbol{s}_{(l)}}$, which is deleted after the first back-propagation of GhostClip and must be re-computed during the second back-propagation. The difference between GhostClip and BK is clearly illustrated via a line-by-line comparison in Appendix C.1. In fact, denoting the total number of model parameters as $M = \sum_l p_{(l)} d_{(l)}$, our trick reduces the time complexity from $10BTM + O(BT^2)$ by Ghost-Clip to $8BTM + O(BT^2)$ according to Table 3. In contrast to Opacus, which book-keeps the per-sample gradients $\boldsymbol{g}_i^{(l)}$ using $O(BM) = O(B \sum_l p_{(l)} d_{(l)})$ memory, we instead book-keep the output gradient with substantially cheaper $O(BT \sum_l p_{(l)})$ memory for small $T$.

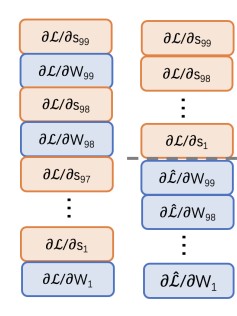

Figure 4: Backward propagation of BK algorithm ($\hat{\mathcal{L}} = \sum_i C_i \mathcal{L}_i$).

**Ghost differentiation trick** This trick improves the time complexity on the first back-propagation in GhostClip, further reducing from $8BTM + O(BT^2)$ to $6BTM + O(BT^2)$ in Table 2. Our idea is to only compute the output gradient $\frac{\partial \mathcal{L}}{\partial \boldsymbol{s}_{(l)}}$ but not the parameter gradient $\frac{\partial \mathcal{L}}{\partial \mathbf{W}}$.

That is, we break the $4BTM$ time complexity of the full back-propagation into two sub-processes, each of $2BTM$ complexity, and remove the unnecessary one.

To be more specific, during the back-propagation of Opacus and GhostClip, the output gradient $\frac{\partial \mathcal{L}}{\partial \boldsymbol{s}}$ and then the parameter gradient $\frac{\partial \mathcal{L}}{\partial \mathbf{W}} = \boldsymbol{a}^\top \frac{\partial \mathcal{L}}{\partial \boldsymbol{s}}$ are computed. However, we can stop after we obtain $\frac{\partial \mathcal{L}}{\partial \boldsymbol{s}}$: we only need the output gradient to compute the clipped parameter gradient $\frac{\partial \sum_i C_i \mathcal{L}_i}{\partial \mathbf{W}}$ in Line 9 of Algorithm 1. Therefore, the ghost differentiation trick sets all parameters to not require gradients (see technical details in Appendix D.2, including the *origin parameter trick* that propagates on a computation graph even when no parameters require gradients).

## 2.2 COMPLEXITY OF DP IMPLEMENTATIONS: A MODULAR ANALYSIS

In this section, we analyze the complexity of DP implementations from their opeartion modules. We summarize the time and space complexity in Table 3 and give the derivation in Appendix B. We will refer to these modules by indices, e.g. ②a for the computation of output gradient.

| Module | ①Forward pass | ②Back-propagation | | ③Ghost norm | ④Per-sample grad instantiation | ⑤Weighted sum of per-sample grad |
| --- | --- | --- | --- | --- | --- | --- |
| | | (a)output gradient | (b)parameter gradient | | | |
| Time complexity | $2BTpd$ | $2BTpd$ | $2BTpd$ | $2BT^2(p+d)$ | $2BTpd$ | $2Bpd$ |
| Space complexity | $pd + BTd$ | $BT(p+d)$ | $pd$ | $2BT^2$ | $Bpd$ | $0$ |

Table 3: Time and space complexity of modules in DP training for one generalized linear layer.

Now we are ready to decompose each implementation, following the flowcharts in Figure 3. Consequently, we can easily write down the complexity of different implementations in Table 2. Such a modular analysis displays the clear effects of the tricks in BK algorithm: the ghost norm trick removes the memory costly ④ from Opacus and FastGradClip; the book-keeping trick removes the ②b in the second back-propagation of FastGradClip and GhostClip; the ghost differentiation trick removes the ②b in the first back-propagation of Opacus and GhostClip.

- Standard (non-DP)= ① + ②a + ②b

- Opacus= ① + ②a + ②b + ④ + ⑤

- FastGradClip= ① + ②a + ④ + ②a + ②b

- GhostClip= ① + ②a + ②b + ③ + ②a + ②b

- BK (ours)= ① + ②a + ③ + ②b

## 2.3 BK IS OPTIMALLY EFFICIENT IN LOW DIMENSION

When the feature dimension $T$ is small, we claim that BK is almost as efficient as the standard non-private training, with a negligible $O(BT^2)$ time and memory overhead by Table 2:

$$\textbf{Memory complexity: } \text{non-DP} \approx \text{BK} \approx \text{GhostClip} < \text{FastGradClip} \ll \text{Opacus}$$
$$\textbf{Time complexity: } \text{non-DP} \approx \text{BK} < \text{FastGradClip} \approx \text{Opacus} < \text{GhostClip}$$

Now, we discuss the cases where the data has low dimension and thus $T$ is small. Generally speaking, the feature dimension $T_{(l)}$ depends on both the data and the model.

For non-sequential input and 1D audio data, $T = 1$. For sequential data such as texts ($T$ being sentence length) or time series ($T$ being time duration), $T_{(l)}$ is fixed across layers. In this case, BK is efficient on short-sequence datasets including GLUE Wang et al. (2019) (e.g. SST2/QNLI/MNLI/QQP) and natural language generation datasets (e.g. E2E/DART), since $T^2 \ll p_{(l)} d_{(l)}$. For instance, Yu et al. (2021); Li et al. (2021); Bu et al. (2022b) applied GPT2 on E2E dataset, which has a sequence length $T \approx 100$ and the number of parameters $p_{(l)} d_{(l)}$ per layer is $2 - 4$M; Yu et al. (2021); Li et al. (2021) applied RoBERTa-large on GLUE datasets, which has a sequence length $T = 256$ and the number of parameters per layer is $1 - 4$M. As illustrated in Figure 5 and Table 1 (extended in Table 9), BK improves the throughput of existing implementations by $25 - 388\%$ on multiple language tasks in Li et al. (2021); Bu et al. (2022b), with minor memory overhead compared to GhostClip and non-private training.

However, on the convolution layers with image data, $T_{(l)}$ is the product of hidden feature sizes (c.f. (Bu et al., 2022a, Section 3)), thus $T_{(l)}$ depends on the original image size and network architecture. For example, larger kernel size/dilation/stride in convolution layer reduces $T_{(l)}$, while larger images have larger $T_{(l)}$ at each layer. Therefore, BK (and GhostClip) may suffer on when training ResNet on ImageNet ($224 \times 224$), as we show in Figure 6 (see also (Bu et al., 2022a, Table 7)), although training the same network efficiently on CIFAR10/100 ($32 \times 32$).

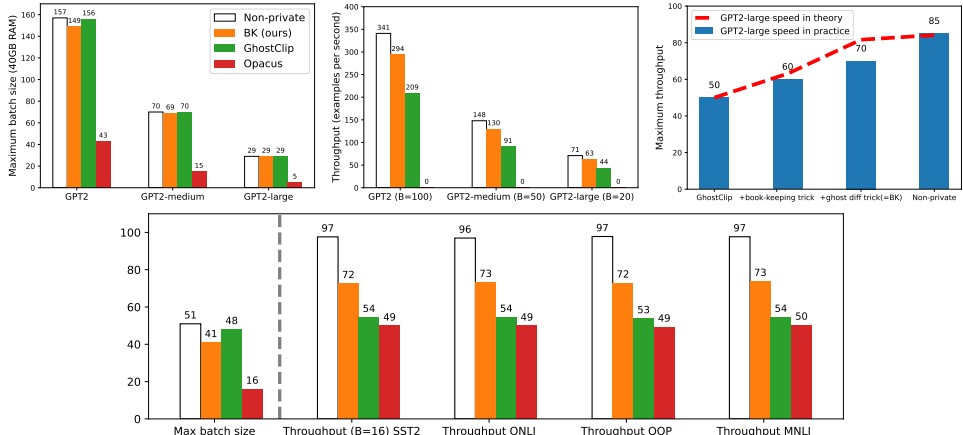

Figure 5: Memory and speed of different DP implementations. Upper: GPT2 on E2E dataset (fixing $B$, DP speed is $0.86 \sim 0.89\times$ of non-DP). Lower: RoBERTa-large on GLUE datasets. Note here the hybrid implementations are equivalent to the base ones, because of the short sequence length.

### 2.4 APPLYING OUR TRICKS TO EXISTING IMPLEMENTATIONS

Our tricks in Section 2.1 can also improve other existing implementations, reducing the time complexity of GhostClip from $10BTpd + 2BT^2(p + d)$ to $6BTpd + 2BT^2(p + d)$, that of Opacus and FastGradClip from $8BTpd$ to $6BTpd$. We highlight that these improved implementations are leveraged to design hybrid implementation in Section 3.2. In addition to DP full fine-tuning, BK is demonstrated in Appendix E.2 to also apply to the parameter efficient fine-tuning like Adapters.

$$\text{GhostClip} = \text{①} + \text{②a} + \text{②b} + \text{③} + \text{②a} + \text{②b} \xrightarrow[\text{book-keeping}]{\text{ghost differentiation}} \text{①} + \text{②a} + \text{③} + \text{②b} \text{ (our BK)}$$

$$\text{Opacus} = \text{①} + \text{②a} + \text{②b} + \text{④} + \text{⑤} \xrightarrow{\text{ghost differentiation}} \text{①} + \text{②a} + \text{④} + \text{⑤}$$

$$\text{FastGradClip} = \text{①} + \text{②a} + \text{④} + \text{②a} + \text{②b} \xrightarrow{\text{book-keeping}} \text{①} + \text{②a} + \text{④} + \text{②b}$$

## 3 HYBRID BOOK-KEEPING: EFFICIENT DP TRAINING IN HIGH DIMENSION

In previous section, we have analyzed DP implementations in the small $T$ regime, where the ghost norm-based GhostClip and BK are efficient. Nevertheless, in the large $T$ and large model regime, none of the base implementations may be efficient (see Figure 6) and we turn to hybrid methods.

### 3.1 LARGE $T$ NECESSITATES NON-GHOST NORM METHOD

A closer look at the space complexity in Table 3 shows that, the ghost norm trick is favored over the per-sample gradient instantiation if and only if $2T_{(l)}^2 < p_{(l)}d_{(l)}$, where $p_{(l)}d_{(l)}$ is the number of parameters at one layer. When this criterion is violated for large $T$, GhostClip/BK can significantly under-perform Opacus/FastGradClip, as shown in Figure 6, Figure 7 and Table 10.

Similar to Section 2.3, we discuss two cases where $T$ is large. For paragraph or document-level language tasks like WikiHop Welbl et al. (2018) and TriviaQA Joshi et al. (2017), $T$ can range from $2000 - 20000$, which makes $2T^2 = 8 - 800$M. For image tasks, particularly on CNN, $T_{(l)}$ varies at each layer with large values on top layers, as the features are less compressed by convolution and pooling. Taking ImageNet and the first convolution layer of VGG11 as an example (Bu et al., 2022a,

Table 3), $2T^2_{(1)} = 5 \times 10^9 \gg p_{(1)}d_{(1)} = 1.7 \times 10^3$. Consequently, ghost norm-based implementations (i.e. GhostClip and BK) costs more than 40GB memory on ResNet18, under $B = 32$, while Opacus only costs 2.5GB. This curse of dimension grows from a difficult issue on ImageNet to an impossible challenge on videos or high-resolution images, e.g. GhostClip cannot train ResNet18 with even one single CelebA-HQ image ($1024 \times 1024$) using a 40GB GPU.

In short, the ghost norm trick is inefficient for large $T$ and the per-sample gradient instantiation is inefficient for large model. Therefore, we must hybridize the base implementations.

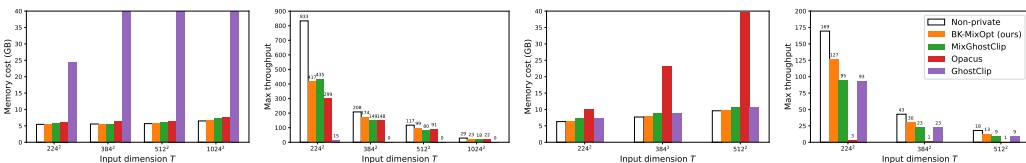

Figure 6: Memory and speed by different implementations on 50000 images. Left: VGG11 (133M;Simonyan & Zisserman (2014)), right is BEiT-large (304M;Bao et al. (2021)). Memory cost uses a physical batch size 1. Throughput uses the maximum physical batch size.

## 3.2 HYBRID IMPLEMENTATIONS VIA LAYERWISE DECISION

We adopt the same layerwise decision as Bu et al. (2022a), known as the mixed ghost norm technique: we use the ghost norm trick on a layer if $2T^2_{(l)} < p_{(l)}d_{(l)}$, and instantiate per-sample gradients otherwise. Therefore, the space complexity of computing the per-sample gradient norm reduces to $\min\{2T^2_{(l)}, p_{(l)}d_{(l)}\}$, which is significantly cheaper than either the ghost norm or the per-sample gradient instantiation in high dimension, as depicted in Table 4 and Figure 7. Consequently, over all layers, the space complexity is lower than both constituting methods, e.g. saving more than $10\times$ memory for the per-sample gradient clipping on ResNet18 (see more models in Table 10).

| | Output size | Space complexity | | | | | |
|---|---|---|---|---|---|---|---|
| | | 18-layer | | 34-layer | | 50-layer | |
| | $H_{\text{out}} \times W_{\text{out}}$ | Ghost norm | Per-sample grad instantiation | Ghost norm | Per-sample grad instantiation | Ghost norm | Per-sample grad instantiation |
| | | $2T^2_{(l)} = 2H^2_{\text{out}}W^2_{\text{out}}$ | $p_{(l)}d_{(l)} = \#$ params | $2T^2_{(l)}$ | $p_{(l)}d_{(l)}$ | $2T^2_{(l)}$ | $p_{(l)}d_{(l)}$ |
| conv1 | $112 \times 112$ | $3.1 \times 10^8$ | $\mathbf{9.4 \times 10^3}$ | $3.1 \times 10^8$ | $\mathbf{9.4 \times 10^3}$ | $3.1 \times 10^8$ | $\mathbf{9.4 \times 10^3}$ |
| conv2_x | $56 \times 56$ | $[2.0 \times 10^7] \times 4$ | $[\mathbf{3.7 \times 10^4}] \times \mathbf{4}$ | $[2.0 \times 10^7] \times 6$ | $[\mathbf{3.7 \times 10^4}] \times \mathbf{6}$ | $[2.0 \times 10^7] \times 9$ | $[\mathbf{4.1 \times 10^3}] \times \mathbf{1}$ $[\mathbf{3.7 \times 10^4}] \times \mathbf{3}$ $[\mathbf{1.6 \times 10^4}] \times \mathbf{5}$ |
| conv3_x | $28 \times 28$ | $[1.2 \times 10^6] \times 4$ | $[\mathbf{7.4 \times 10^4}] \times \mathbf{1}$ $[\mathbf{1.5 \times 10^5}] \times \mathbf{3}$ | $[1.2 \times 10^6] \times 8$ | $[\mathbf{7.4 \times 10^4}] \times \mathbf{1}$ $[\mathbf{1.5 \times 10^5}] \times \mathbf{7}$ | $[2.0 \times 10^7] \times 1$ $[1.2 \times 10^6] \times 11$ | $[\mathbf{3.3 \times 10^4}] \times \mathbf{1}$ $[\mathbf{6.6 \times 10^4}] \times \mathbf{7}$ $[\mathbf{1.5 \times 10^5}] \times \mathbf{4}$ |
| conv4_x | $14 \times 14$ | $[\mathbf{7.7 \times 10^4}] \times \mathbf{4}$ | $[2.9 \times 10^5] \times 1$ $[5.9 \times 10^5] \times 3$ | $[\mathbf{7.7 \times 10^4}] \times \mathbf{12}$ | $[2.6 \times 10^5] \times 1$ $[5.9 \times 10^5] \times 11$ | $[1.2 \times 10^6] \times 1$ $[\mathbf{7.7 \times 10^4}] \times \mathbf{17}$ | $[\mathbf{1.3 \times 10^5}] \times \mathbf{1}$ $[2.6 \times 10^5] \times 11$ $[5.9 \times 10^5] \times 6$ |
| conv5_x | $7 \times 7$ | $[\mathbf{4.8 \times 10^3}] \times \mathbf{4}$ | $[1.2 \times 10^6] \times 1$ $[2.4 \times 10^6] \times 3$ | $[\mathbf{4.8 \times 10^3}] \times \mathbf{6}$ | $[1.2 \times 10^6] \times 1$ $[2.4 \times 10^6] \times 5$ | $[\mathbf{4.8 \times 10^3}] \times \mathbf{9}$ | $[5.2 \times 10^5] \times 1$ $[1.0 \times 10^6] \times 5$ $[2.4 \times 10^6] \times 3$ |
| linear | $1 \times 1$ | $\mathbf{2}$ | $5.1 \times 10^5$ | $\mathbf{2}$ | $5.1 \times 10^5$ | $\mathbf{2}$ | $2.0 \times 10^6$ |
| Total complexity | | 399M | 11.5M | 444M | 21.6M | 528M | 22.7M |
| Complexity by mixed ghost norm | | 1.0M | | 2.3M | | 2.8M | |

Table 4: Space complexity of the per-sample gradient clipping (not the entire DP algorithm) for $B = 1$ on ImageNet $224 \times 224$. Layerwise decision of hybrid BK algorithms is highlighted in bold.

In contrast to the mixed ghost clipping (MixGhostClip) in Bu et al. (2022a), which hybridizes Fast-GradClip and GhostClip, we boost the efficiency by hybridizing our BK with the improved Fast-GradClip/Opacus in Section 2.4. We propose BK-MixOpt (and BK-MixGhostClip as an intermediate product only for comparison) and use MixGhostClip as a reference point,

- MixGhostClip = ① + ②a + ②b + $\min\{$③,④$\}$ + ②a + ②b $\approx \min\{$GhostClip, FastGradClip$\}$,

- BK-MixGhostClip = ① + ②a + $\min\{$③,④$\}$ + ②b $= \min\{$BK, improved FastGradClip$\}$,

- BK-MixOpt = ① + ②a + $\min\{$③ + ②b, ④ + ⑤$\}$ $= \min\{$BK, improved Opacus$\}$.

The hybrid BK algorithms are presented in Algorithm 5. We summarize the layerwise complexity in Table 5, from which we derive the overall complexity in Table 8 and observe that BK has almost the same complexity as non-DP training. Note that in low dimension, the mixed ghost norm is equivalent to the ghost norm, hence MixGhostClip/BK-MixOpt is equivalent to GhostClip/BK, respectively.

| Method | Type | Modification to previous variant | Time complexity | Space complexity |
|---|---|---|---|---|
| Non-DP | | | $6BTpd$ | $pd + 3BTd + BTp$ |
| Opacus | | Instantiate per-sample gradient | $8BTpd$ | $Bpd + 3BTd + BTp$ |
| FastGradClip | base | Not store per-sample gradient using a second back-propagation | $8BTpd$ | $Bpd + 2BTd + BTp$ |
| GhostClip | | Not instantiate per-sample gradient using ghost norm trick | $10BTpd + 2BT^2(p + d)$ | $2BT^2 + 3BTd + BTp$ |
| BK (ours) | | Simplify the two back-propagations | $6BTpd + 2BT^2(p + d)$ | $2BT^2 + 3BTd + BTp$ |
| MixGhostClip | hybrid | Mix ways to compute grad norm | $8BTpd + \langle 2BTpd, 2BT^2(p + d)\rangle$ | $\min\{2BT^2, Bpd\} + 3BTd + BTp$ |
| BK-MixGhostClip | | | $6BTpd + \langle 2BTpd, 2BT^2(p + d)\rangle$ | $\min\{2BT^2, Bpd\} + 3BTd + BTp$ |
| BK-MixOpt | | Mix ways to compute weighted grad | $6BTpd + \langle 0, 2BT^2(p + d)\rangle$ | $\min\{2BT^2, Bpd\} + 3BTd + BTp$ |

Table 5: Complexity of DP implementations on one layer. Here $\langle\rangle$ means between two values. The time complexity of BK-MixOpt is $6BTpd + 2BT^2(p + d) \cdot \mathbb{I}\{2T^2 < pd\}$.

### 3.3 EFFECT OF MODEL ARCHITECTURE & FEATURE DIMENSION ON HYBRIDIZATION

We dive deeper to understand when the hybridization favors the ghost or non-ghost norm tricks.

From a model architecture viewpoint, transformers such as ViT, RoBERTa, GPT tend to prefer the ghost norm: for moderate-sequence text data and moderate-dimension image data, hybrid BK algorithms are close or equivalent to the base BK algorithm (see right-most plot in Figure 7). However, CNN prefers the per-sample gradient instantiation at top layers, and there exists a depth threshold below which the ghost norm is more efficient. Hence the hybridization is necessary to take advantages of both worlds. From the feature dimension viewpoint, larger input means this depth threshold is deeper, e.g. from the 9-th layer of ResNet18 to the 17-th layer in Figure 7, when the image size increases from $224 \times 224$ to $512 \times 512$. We visualize this effect of feature dimension on various models in Appendix G.

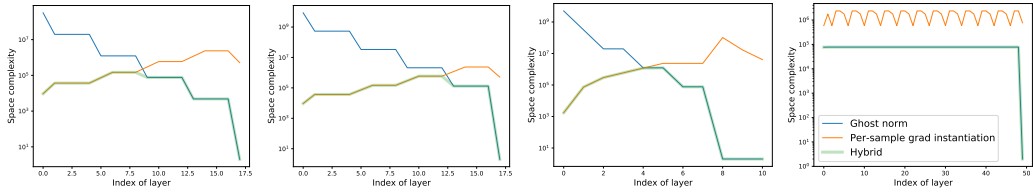

Figure 7: Layerwise space complexity of computing the per-sample gradient norm. Left to right: ResNet18 ($224 \times 224$), ResNet18 ($512 \times 512$), VGG11 ($224 \times 224$), and ViT-base ($224 \times 224$).

## 4 DISCUSSION

In this work, we propose the Book-Keeping (BK) algorithms to effciently implement DP optimizers using three tricks: ghost norm, book-keeping, and ghost differentiation. Our BK reduces the time and space complexity of DP training to the similar level of the standard training. Specially, we develop hybrid BK to overcome the computational challenge of training large models with high-dimensional data, and we extend BK to parameter efficient fine-tuning such as LoRA and Adapter.

One minor limitation of this work is that BK (and GhostClip) only applies to the weights, not the biases, of the generalized linear layers, i.e. embedding, linear, and convolution layers, though these weights constitute 99.9% of the trainable parameters (see Table 7). Implementation-wise, although BK should be as fast as the standard training for small $T$, e.g. on MLP where $T = 1$, we observe some gap between the theoretical complexity and the throughput in practice. This gap is mainly due to the mechanism of Pytorch hooks which can be possibly optimized by customizing the CUDA kernel or using the symbolic programming.

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
