# OpenReview forum: "Differentially Private Optimization on Large Model at Small Cost"
_ICLR.cc/2023/Conference — Submitted to ICLR 2023_

### Official Review · Reviewer_cPLw · 2022-10-19

**Confidence:** 5
**Correctness:** 2
**Technical Novelty And Significance:** 2
**Empirical Novelty And Significance:** 2
**Recommendation:** 5

**Clarity, Quality, Novelty And Reproducibility:**

quality: the paper proposes an interesting modification for attaining speed ups, but there are several factually incorrect claims.
clarity: the paper could improve on the writing and presentation. for instance, authors don't distinguish between \cite and \citep.
novelty: the minor trick proposed by the paper is new, to the best of my knowledge.
reproducible: results are likely reproducible to the best of my knowledge. authors agree to release code, which facilitates reproducibility.

**Strength And Weaknesses:**

strength: the authors show empirical gains for common setups of DP training in terms of wall-clock time compared to previous approaches.

weakness: the paper has several weaknesses, some of which are technical and the others are related to presentation.

technical weaknesses:

while the idea for modifying FastGradClip may seem natural, it's worthwhile to note that the current presentation is heavily based on a particular framework -- PyTorch. for instance, Algorithm 1 explicitly states the procedure quoting PyTorch forward and backward hooks. i believe the high-level idea is applicable in other frameworks, but the current algorithmic sol'n is limiting and isn't designed based on shared primitives of automatic-differentiation framework.

in the main text and appendix C, the authors argue that one major difference between BK and GhostClip is that BK doesn't compute non-private gradients. note that GhostClip need not compute non-private gradients at all, since this quantity is not used anywhere. in fact, computing the non-private gradient is likely an artifact of a particular implementation and is not a general drawback of the approach (of course, neither is this a drawback / limitation of FastGradClip). by not considering this point, the authors likely have "inflated" their computational advantages.

the authors make several factually incorrect claims. for instance on page 4, authors say "For instance, Li et al. (2021) shows that, when training GPT2-large (774M parameters), Opacus Yousefpour et al. (2021) and JAX Subramani et al. (2021) cannot fit even one single sample into a 16GB GPU."
  - note the original paper by Li et al. does not present profile results with GPUs which have 16GB video RAM. the selected GPU is TITAN RTX which has 24GB video RAM.

**Summary Of The Paper:**

the paper proposes a (pytorch-based)  implementation for summing clipped per-sample gradients in DP-SGD and demonstrates empirical gains in terms of compute time.

**Summary Of The Review:**

the paper proposes a (pytorch-based)  implementation for summing clipped per-sample gradients in DP-SGD and demonstrates empirical gains in terms of compute time. the trick is new to the best of my knowledge, but the paper has several technical issues.

---

> ### Author Response · Authors · 2022-11-12
> **Response**
>
> We respond to each comment of the reviewer and will modify typos (including the cite/citep issues) in the camera-ready version. Please consider raise the score if you are happy with our response.
>
> **Comment:** *Strength: the authors show empirical gains for common setups of DP training in terms of wall-clock time compared to previous approaches.*
>
> **Response:** We thank the reviewer for appreciating our improvement on the speed, which is highlighted in the revised Table 1: we complement our complexity analysis to show that our BK algorithm is the first DP algorithm to achieve time complexity about 90\% of non-DP training.
>
> We would also like to point out our contribution on the memory-saving in a generally applicable manner (e.g. Opacus cannot fit a single sample with GPT2-large; GhostClip cannot fit a single 512*512 image with VGG11).
>
> **Comment:** *while the idea for modifying FastGradClip may seem natural, it's worthwhile to note that the current presentation is heavily based on a particular framework -- PyTorch. for instance, Algorithm 1 explicitly states the procedure quoting PyTorch forward and backward hooks. i believe the high-level idea is applicable in other frameworks, but the current algorithmic sol'n is limiting and isn't designed based on shared primitives of automatic-differentiation framework.*
>
> **Response:** We agree that our BK algorithm is high-level and extendable to other platforms like Tensorflow or JAX. We have described the choice of platforms in "Additionally, these methods can be compiled on different platforms (part IV) such as Tensorflow
> 2(XLA), JAX and Pytorch", above Section 1.1.
>
> Therefore, our algorithm is **NOT limiting** and indeed "designed based on shared primitives of automatic-differentiation framework". We visualize our algorithm in Figure 3 and the newly added Figure 4, which clearly demonstrate the case since all back-propagation operations, regardless of the platforms, compute the output gradient and the parameter gradient. That is, these computations are shared across platforms and BK is essentially an elegant re-ordering of them by Figure 4. Due to the page limit, the discussion of auto-differentiation is left in Appendix B.
>
> **Comment:** *in the main text and appendix C, the authors argue that one major difference between BK and GhostClip is that BK doesn't compute non-private gradients. note that GhostClip need not compute non-private gradients at all, since this quantity is not used anywhere. in fact, computing the non-private gradient is likely an artifact of a particular implementation and is not a general drawback of the approach (of course, neither is this a drawback / limitation of FastGradClip). by not considering this point, the authors likely have "inflated" their computational advantages.*
>
> **Response:** We agree that the non-private gradient in Line 8 of Algorithm 2 is not used anywhere and theoretically should not be computed in GhostClip as well as Opacus, to save time. However, this cannot be simply removed from GhostClip nor Opacus in practice, because the back-propagation will not launch if none of the parameters require gradients. From this viewpoint, computing the non-private gradient is a drawback that we have to solve by our ghost differentiation with **origin parameter trick** in Appendix D.3, where only the first parameter (close to input) requires the gradient.
>
> To assure the reviewer about our contribution, we add a new top-right subplot in Figure 5, which demonstrates a 20\% speedup on GPT2-large if we only apply the book-keeping trick but still compute the non-private like existing methods. We emphasize that this improvement has been discussed in the original submission from a theoretical perspective: we claimed 8/6=1.333X improvement in time complexity, see page 5 'Book-keeping trick' paragraph: "our trick reduces the time complexity from $10BTM + O(BT^2)$ by GhostClip to $8BTM + O(BT^2)$ according to Table 3."
>
> **Comment:** *the authors make several factually incorrect claims. for instance on page 4, authors say "For instance, Li et al. (2021) shows that, when training GPT2-large (774M parameters), Opacus Yousefpour et al. (2021) and JAX Subramani et al. (2021) cannot fit even one single sample into a 16GB GPU." Note the original paper by Li et al. does not present profile results with GPUs which have 16GB video RAM. the selected GPU is TITAN RTX which has 24GB video RAM.*
>
> **Response:** We thank the reviewer for spotting this and we have changed 16 to 24 in the revision. We are grateful if the reviewer can let us know other factually incorrect claims besides this single one (and this typo should not compromise our main contribution).

---

### Official Review · Reviewer_ac6D · 2022-10-23

**Confidence:** 3
**Correctness:** 3
**Technical Novelty And Significance:** 3
**Empirical Novelty And Significance:** Not applicable
**Recommendation:** 6

**Clarity, Quality, Novelty And Reproducibility:**

The paper is very clearly written and comprehensive. There are not a lot of experimental results in the paper, but they don't seem needed.


**Strength And Weaknesses:**

The paper explains the implementation very well, comparing nicely against various previous algorithms. It also describe the limitations and the resulting hybrid algorithm well.

The authors mention two tricks, but it was not easy to distinguish these.  Q: What is the purpose of Line 8 in Algorithm 2? The output is not used anywhere else in the code. This is mentioned as one of the two important tricks in the paper, and adds considerably to the time and memory requirements of GhostClip.

The main novelty is in the implementation of the tricks --- the trick itself is straightforward --- instead of recomputing the gradients, the authors store them from the previous pass.

In equation 2, you multiply 2 vectors --- should this be a dot product?

**Summary Of The Paper:**

The paper proposes a new trick to improve the runtime of DP training with GhostClip. One large cost in DP training is the computation of the per example gradient, which must be clipped before being averaged, so that no example can have an outsized influence on the model. The authors enhance a previously known trick to reduce this cost.

GhostClip, proposed in 2015, allows the computation of the norms of the per example gradients efficiently (i.e. without storing the per-example gradient). Then GhostClip uses a second backpropagation pass to compute the gradients of the weighted loss --- the weighted loss is the sum of the loss for each example weighted by the norm of its gradient, thus achieving clipping.

In the current paper, the authors propose to store the per example gradients of the outputs in the first pass, while computing the norms of the per example gradients. These are used to recompute the gradient of the weighted loss, without needing to use a second backprop pass. Since only the per example gradients of the outputs are stored, and not per example gradients of the parameters, this memory is substantially less than the memory used in standard DP implementations.

The paper then points out that the GhostClip trick works well only when the number of features in a layer is small (each layer's input is of size batchsize * features * channels). For layers where the number of features is large (e.g. the initial layers of a ResNet for ImageNet), the authors do not use GhostClip, instead keeping per example gradients of the parameters. Thus they are able to reduce the huge memory requirement of GhostClip for these cases.

**Summary Of The Review:**

The paper seems to me to be very clear and useful. My only concern is with the question above, hopefully the authors can address it in the discussion.

---

> ### Author Response · Authors · 2022-11-12
> **Response**
>
> We thank the reviewer for the deep understanding of this paper and the discussion of technical details. In the revised submission Table 1, we complement our complexity analysis to show that our BK algorithm is the first DP algorithm to achieve time complexity about 89-97\% of non-DP training.
>
> **Comment:** *The authors mention two tricks, but it was not easy to distinguish these. Q: What is the purpose of Line 8 in Algorithm 2? The output is not used anywhere else in the code. This is mentioned as one of the two important tricks in the paper, and adds considerably to the time and memory requirements of GhostClip.*
>
> **Response:** We agree that the non-private gradient in Line 8 of Algorithm 2 is not used anywhere and theoretically should not be computed in GhostClip as well as Opacus, to save time. However, we discover that this cannot be simply removed from GhostClip nor Opacus in practice, because the back-propagation will not launch if none of the parameters require gradients. This is the motivation of ghost differentiation with **origin parameter trick** in Appendix D.3, where only the first parameter (close to input) requires the gradient.
>
> To distinguish the ghost differentiation trick and the book-keeping trick, we refer the reviewer to page 5. The ghost differentiation removes the parameter gradient (operation 2b) in the first back-propagation of GhostClip, while the book-keeping removes the output gradient (operation 2a) in the second back-propagation. We add Figure 4 to demonstrate the final pipeline when both new tricks are applied. We also add the top-right subplot in Figure 5 to demonstrate the separate effects of our tricks.
>
> **Comment:** *In equation 2, you multiply 2 vectors --- should this be a dot product?*
>
> **Response:** Yes. We have modified in the revision.

---

### Official Review · Reviewer_QN3z · 2022-10-24

**Confidence:** 4
**Correctness:** 4
**Technical Novelty And Significance:** 2
**Empirical Novelty And Significance:** 2
**Recommendation:** 5

**Clarity, Quality, Novelty And Reproducibility:**

The organization of the paper has some problems. A few examples:

1. Table 1 is shown in Section 1.1 but the first explanation of it lies in Section 2.3.

2. Figure 3 is referred to before Figure 2, but it appears much later than Figure 2.

**Strength And Weaknesses:**

Strength

1. Table 2 provides a clear and comprehensive comparison of the existing methods.

2. The paper provides a complete comparison of pseudo-codes for their algorithm and the competitors.

Weaknesses

W1. The paper combines some tricks from existing works to save computational costs of deep learning with DP. However, when comparing their code with SOTA, it seems that the proposed algorithm only replaces one summation step with matrix operations, which lacks novelty.

W2. The paper repeatedly emphasizes that it can reduce the time and memory complexity of DP deep learning. However, the actual reduction is only a constant ("1.24× training speed in practice").  Moreover, according to the experiments in Figures 2 and 5, when the input dimension is large, the proposed algorithm only shows minor improvement in time or memory compared to the state of the art.

W3. Figure 3 is difficult to read.

W4. The paper is not self-contained in the sense that it refers to a number of tables in the main text, without mentioning that those tables are only available in the supplementary material.

**Summary Of The Paper:**

This paper implements existing DP optimizers with 1.24x training speed than the most efficient competitor. Previous methods accelerate the DP training speed by using two rounds of back-propagation or sacrificing memory. This paper uses one back-propagation and never instantiates per-sample gradients so that it reduces computational costs.

**Summary Of The Review:**

The novelty of the paper seems low, and the propose solution only achieves minor improvements against the state of the art.

---

> ### Author Response · Authors · 2022-11-12
> **Response**
>
> We respond to each comment of the reviewer and will modify typos in the camera-ready version. Please consider raise the score if you are happy with our response.
>
> **Comment:**
> *The paper combines some tricks from existing works to save computational costs of deep learning with DP. However, when comparing their code with SOTA, it seems that the proposed algorithm only replaces one summation step with matrix operations, which lacks novelty.*
>
> **Response:** We thank the reviewer for the comments. Our Book-Keeping algorithm is indeed a unified algorithm, leveraging **novel tricks** (e.g. **ghost differentiation and book-keeping output gradient**) as well as existing tricks (e.g. ghost norm and per-sample gradient instantiation). Specifically, our novel tricks are the key to improve the time complexity from 10BTM of GhostClip to 6BTM. This is clearly demonstrated in Table 2, as none of the previous SOTA algorithms can be uniformly suitable for all tasks like our BK-MixOpt.
>
> To address your concern in "However, when comparing their code with SOTA, it seems that the proposed algorithm only replaces one summation step with matrix operations", we refer the reviewer to the line by line comparison in Appendix C. The replacement is indeed novel and highly insightful, reducing the complexity of GhostClip from 10BTM to 6BTM plus small extra cost, and the contribution should not be under-estimated due to its simplicity. To elaborate on the novelty in the book-keeping, we add Figure 4 and upper-right subplot in Figure 5, to visualize the elegant pipeline of our BK algorithm.
>
>
> **Comment:**
> *The paper repeatedly emphasizes that it can reduce the time and memory complexity of DP deep learning. However, the actual reduction is only a constant ("1.24× training speed in practice"). Moreover, according to the experiments in Figures 2 and 5, when the input dimension is large, the proposed algorithm only shows minor improvement in time or memory compared to the state of the art.*
>
> **Response:** We thank the reviewer for the consideration of performance. Note that in the revised submission Table 1, we complement our complexity analysis to show that our BK algorithm is the first DP algorithm to achieve time complexity about 89-97\% of non-DP training. Firstly, our performance gain is both theoretical by 1.666× and empirical by $1.33×-\infty$ times (e.g. second subplot in Figure 6) over GhostClip, as well as by 1.96-50× over Opacus. Here the gap between theoretical and empirical improvement is due to the known issue of Pytorch hook that is slower than the C++ backend which implements the non-DP back-propagation.
>
> In the original submission, 1.24× was the minimal improvement so it would be unfair to evaluate our contribution based on the worst case. This number has been significantly improved to 1.36× after we correct a performance blocker in the code.
>
> Secondly, in "Figure 2 and 5, when the input dimension is large", the improvement over speed is still large: we have about 1.4× speedup on wide networks in Figure 2 and on RoBERTa/GPT2 in Figure 5.
>
>
> **Comment:** *Figure 3 is difficult to read.*
>
> **Response:** We thank the reviewer for this comment. We agree that the figure is slightly squeezed but it is necessary to demonstrate the tricks we invented in this work or extracted from previous methods. We also add a new Figure 4 to clarify our BK algorithm. We hope this would enhance the readability.

---

### Decision · Program_Chairs · 2023-01-20

**Decision:**

Reject

**Justification For Why Not Higher Score:**

see review

**Justification For Why Not Lower Score:**

N/A

**Metareview: Summary, Strengths And Weaknesses:**

This work presents two "tricks" to speed up the computation of clipped gradients in differentially private training. These are useful engineering insights that give meaningful, even if relatively modest, speed-ups. At the same time the overall novelty and significance of these results is not sufficiently high for acceptance.